# Modular and Portable System Design for 3D Imaging of Breast Tumors Using Electrical Impedance Tomography

**DOI:** 10.3390/s24196370

**Published:** 2024-09-30

**Authors:** Juan Carlos Gómez Cortés, José Javier Diaz Carmona, Alejandro Israel Barranco Gutiérrez, José Alfredo Padilla Medina, Adán Antonio Alonso Ramírez, Joel Artemio Morales Viscaya, J. Jesús Villegas-Saucillo, Juan Prado Olivarez

**Affiliations:** Departamento de Ingeniería Eléctrica y Electrónica, Tecnológico Nacional de México en Celaya, A. García-Cubas No. 600 Pte. Esq. Av. Tecnológico, Col. Alfredo V. Bonfil, Celaya C.P. Guanajuato 38010, Mexico; m1703042@itcelaya.edu.mx (J.C.G.C.); javier.diaz@itcelaya.edu.mx (J.J.D.C.); israel.barranco@itcelaya.edu.mx (A.I.B.G.); alfredo.padilla@itcelaya.edu.mx (J.A.P.M.); d2203002@itcelaya.edu.mx (A.A.A.R.); d2003026@itcelaya.edu.mx (J.A.M.V.); jesus.villegas@itcelaya.edu.mx (J.J.V.-S.)

**Keywords:** 3D electrical impedance tomography, tumor detection, health technology

## Abstract

This paper presents a prototype of a portable and modular electrical impedance tomography (EIT) system for breast tumor detection. The proposed system uses MATLAB to generate three-dimensional representations of breast tissue. The modular architecture of the system allows for flexible customization and scalability. It consists of several interconnected modules. Each module can be easily replaced or upgraded, facilitating system maintenance and future enhancements. Testing of the prototype has shown promising results in preliminary screening based on experimental studies. Agar models were used for the experimental stage of this project. The 3D representations provide clinicians with valuable information for accurate diagnosis and treatment planning. Further research and refinement of the system is warranted to validate its performance in future clinical trials.

## 1. Introduction

Breast cancer remains the most common malignancy among women globally, characterized by the uncontrolled growth of cells in the breast tissue [1]. Tumors can be classified into benign or malignant, with the latter having the potential to invade surrounding tissues and metastasize [2]. Early detection of malignant tumors is crucial for improving treatment outcomes and survival rates [3]. Conventional imaging methods, such as mammography and MRI, are widely used for breast cancer screening but often have limitations, including high cost, radiation exposure, and limited accessibility in remote areas [4].

The prevalence and impact of breast cancer make it a significant global health concern, necessitating ongoing research and advancements in diagnosis and treatment strategies [5]. Early diagnosis often facilitates more effective and less invasive treatment options.

Globally, there are significant inequalities in access to medical care and resources for breast cancer diagnosis and treatment. Breast cancer poses a paradoxical challenge as its incidence is lower among socioeconomically disadvantaged women; however, a disproportionate number of them succumb to the disease [6]. Research highlights significant disparities in the utilization of breast cancer screening, with lower uptake observed among women from poorer households and those with limited education [7]. Furthermore, the spatial distribution of advanced-stage breast cancer diagnoses reveals a correlation with inequalities related to local income [8]. These findings underscore the complexity of breast cancer detection inequalities, urging a comprehensive approach to address socio-economic and geographical factors that contribute to disparate outcomes.

In recent years, Electrical Impedance Tomography (EIT) has emerged as a noninvasive, radiation-free, and low-cost alternative for medical imaging [9]. EIT works by measuring the electrical impedance distribution within biological tissues, exploiting the fact that malignant tumors typically exhibit different electrical properties than healthy tissue [10,11,12,13]. Despite its potential, EIT is still under development, particularly in the context of breast cancer detection, where limitations in resolution and patient testing have hindered its widespread adoption [14]. However, EIT also has limitations, such as the need to assume simplified tissue models and its relatively lower resolution compared to more established medical imaging techniques. Despite this, EIT demonstrates significant potential when combined with other imaging modalities. Current research has explored its applications in monitoring lung function, detecting blood flow changes, assessing brain activity, and identifying soft tissue abnormalities [15,16]. It has also been used to monitor ventilation during anesthesia and in intensive care units [17]. The electrical properties of tissues, crucial for EIT, have been characterized through ex-vivo testing, revealing lower impedance values in tumors compared to healthy tissue [18,19].

A major challenge for EIT in breast cancer research is the lack of patient testing in many tumor detection studies, which makes direct comparison with widely used techniques, such as mammography, difficult [20]. Several studies have explored the application of EIT for breast cancer detection due to its noninvasive and cost-effective nature. While previous works have shown EIT’s potential in identifying tumors, challenges remain in resolution, scalability, and practical clinical implementation. For instance, [11] developed an EIT system for tumor detection but faced limitations in spatial resolution, making it difficult to detect small tumors accurately. Similarly, [21] integrated EIT with other imaging techniques to improve detection accuracy, though the complexity and high costs of such systems limit their applicability in resource-constrained environments.

In contrast, our proposed system addresses these gaps by introducing a portable and modular EIT platform specifically designed for breast tumor detection. Unlike prior systems, our modular design allows for easy customization and future upgrades, which is essential for adapting the technology to diverse clinical needs. Additionally, the focus on portability enables the system to be deployed in remote or underserved areas, offering a viable alternative to conventional imaging techniques such as mammography and MRI.

Compared to other noninvasive imaging methods, such as microwave imaging—widely studied for breast tumor detection due to its ability to differentiate healthy and malignant tissues based on dielectric properties—EIT has distinct advantages. Although microwave imaging provides high-contrast images, it often faces challenges with equipment complexity and signal distortion caused by breast tissue heterogeneity [22,23]. In contrast, EIT systems, including ours, offer simpler hardware setups, lower operating costs, and avoid the high-energy radiation associated with microwave techniques. EIT’s portability and lower energy requirements also make it more accessible for point-of-care applications, particularly in resource-limited settings [24].

While microwave imaging has shown promise in terms of sensitivity and specificity for detecting malignant tumors, our system’s modular and portable design allows for rapid adjustments and potential integration with other techniques to improve detection accuracy. Furthermore, the 3D imaging capabilities of our system enable precise tumor localization within breast tissue, addressing a common challenge in microwave imaging due to signal interference.

We validated the performance of our proposed system through experimental tests with agar-based breast models designed to mimic the electrical properties of tumors. These tests indicate that our system provides improved accuracy in tumor localization, particularly in identifying cross-point coordinates, compared to previous EIT-based systems. These promising results highlight the potential of our system to bridge existing gaps in breast cancer screening technologies, especially in resource-limited environments.

Our prototype consists of a portable EIT system for breast tumor detection featuring a ring-shaped array of electrodes mounted on a wearable brassiere cup. The system’s modular design facilitates easy upgrades, modifications, and maintenance. During the experimental phase, we used agar models with tumor emulators (e-tumors) that simulate the electrical properties of real tumors. These emulated tumors, made from materials like agar-agar, are commonly used to evaluate the accuracy and sensitivity of imaging techniques such as EIT [25].

The primary contribution of this work is the development of a cost-effective, portable EIT-based imaging system for breast tumor detection, which addresses key challenges in accessibility and scalability compared to existing methods. Our results demonstrate the system’s potential for preliminary screening, with promising implications for early breast cancer diagnosis, particularly in resource-limited settings.

## 2. Materials and Methods

### 2.1. Control System

The first step was to simulate the electrical impedance tomography technique to understand its potential and limitations. The positive results obtained during the simulation phase, which demonstrated the feasibility and effectiveness of EIT in imaging [26], provided the impetus for the subsequent creation of a physical prototype. The purpose of the prototype was to translate the promising simulation results into a tangible, real-world application. This transition from simulation to prototype represents a critical step in the validation and practical implementation of EIT technology.

This system is designed with a modular approach that allows for adaptation and customization to different medical testing environments and easy upgrades with interchangeable modules. The compact and portable design allows for use in remote locations. The proposed system, shown in Figure 1, outlines a tumor detection setup in agar models. This system has been designed with a modular approach to allow adaptation and customization to different medical testing environments and easy upgrades with interchangeable modules. The compact and portable nature of the device allows it to be used in remote locations. The proposed system, shown in Figure 1, outlines a setup for tumor detection in agar models.

### 2.2. Latex Cup Electrode Distribution

For current injection into the agar models and impedance measurement, AgCl electrodes with a diameter of 10 mm were used. The electrodes are distributed in a latex insulating cup with dimensions similar to the standard B-cup, as shown in Figure 2. A side view of the latex cup Figure 2a is presented; the electrodes were distributed in three levels of 8 electrodes at each level. A total of 24 electrodes were used; they were distributed in 3 rings of 8 electrodes each; in each ring, each electrode was separated 45° from the adjoining electrodes Figure 2b.

### 2.3. Current Injection Distribution

A multiplexer was used to distribute the injection current between the electrodes. The multiplexer (MUX) is an electronic device used to combine multiple signals on a single line or to select one of several signals and transmit it on a single output line [27]. Basically, it acts as a “selector” to direct a specific signal from among several to a desired location. The MUX allows the injection current to be distributed to all the electrodes until the sweep of the electrode pairs is complete [28]; the impedance meter sends the injection current and performs the measurement of the impedance values. Figure 3 shows the diagram explaining the relationship of the MUX to the other components of the system. Each multiplexer has 16 output channels (C) and requires 4-bit logic combinations (S) to select the output channel of the impedance meter signal (SIG). The MUX outputs are connected to the ring electrode array of the latex cup.

A state machine is used to control the data acquisition process and the switching between different electrodes during scanning, optimizing data transfer and reducing the complexity of data acquisition [29]. The state machine manages the order in which the electrodes are activated during measurement acquisition. It instructs the MUX to sequentially select each pair of electrodes to send the electric current and measure the resulting voltage. The diagram in Figure 4 shows the connection between the state machine, the MUX, and the impedance meter.

### 2.4. Impedance Meter

The impedance meter has a measuring frequency of 4 Hz to 8 MHz. For the experimental stage of this project, ranges from 10 kHz to 100 kHz were used with an injection current of 5 mA to continue with the safe values reported by previous research [20]. The impedance meter receives a start signal from the computer, the impedance meter performs the measurement, and sends an End of Measurement (EOM) signal to the state machine; the state machine switches to the next state and sends a logic combination to the MUX to activate the next combination of injector electrodes on the sample. The impedance meter sends an EOM signal to change the output logic combination in the state machine until the sweep is complete; at the end of all combinations, the state machine returns to its initial state and waits for the start signal from the computer. A total of 6 MUXs were used to measure the 48 electrodes of the system (24 electrodes for each breast). The state machine was implemented using an Arduino mega 2560. A double-layer PCB (Printed Circuit Board) was designed and built to integrate the state machine (Arduino 1.8.13) and the multiplexers on the same board. The dimensions of this PCB are 100 mm by 100 mm. Figure 5 shows the physical appearance of the designed PCB. The PCB was designed using the software Altium 21.4.1 to optimize the size and use of components within a single double-layer PCB in an effort to maintain a standard size (100 mm × 100 mm) for PCB board manufacturing without losing the advantages of a modular system (easy replacement and upgrade of components). Figure 5a shows the bottom layer of the PCB, which communicates the multiplexers with the signals coming from the impedance meter and the state machine. The top layer of the PCB (Figure 5b) contains the Arduino mega 2560 board that functions as the control machine, which allows modifications and adjustments when testing. For example, modifications of a number of measurement injection combinations between electrodes can be made.

### 2.5. Wearable Brassiere Cup Design

A modified brassiere cup device that could be worn by the patient was proposed to increase the portability of the system. A standard B-cup brassiere was designed with a Velcro liner that serves to attach the latex cup to the electrodes, following the distribution of electrodes described in Figure 2. Figure 6 shows the brassiere cup with the 48 electrodes attached to the latex cups.

Figure 7 shows the complete physical prototype, with all modules interconnected for impedance measurement. The impedance meter is delimited with red dotted frame, the control module has blue dotted frame, and the brassiere cup has green dotted frame.

### 2.6. Agar Models

Agar-based models are widely used for validation purposes in electrical impedance tomography (EIT) projects. These models, typically composed of agar gel, simulate the electrical properties of biological tissues and provide a controlled environment for testing and refining EIT techniques.

Agar is chosen for its biocompatibility and stability, ensuring consistent and reliable results in breast imaging and tumor localization experiments. Agar models mimic the conductivity of biological tissues, allowing researchers to evaluate the accuracy of impedance measurements under conditions similar to the human body [25,30].

For the experimental stage, the conductivity of the agar models was modified. Bennett’s equation is a fundamental component in the field of electrical impedance tomography (EIT). It describes the electrical impedance of brain surfaces and is often referenced in studies of neural tissue conductivity [31]. Although the exact formulation of Bennett’s equation may vary in different contexts, it generally includes mainly electrical impedance. In the context of using agar models for breast simulation, equation 1 has been implemented. This variation, proposed and validated by Gonzalez [32], allows the conductivity values to be modified by varying the values of NaCl and deionized water in the agar preparation.
σ = 215 × (gn/tv) + 0.0529(1)
where “σ” is the conductivity expressed in Siemens/meter, “gn” is the grams of NaCl in the sample, and “tv” represents the total volume of the mixture expressed in milliliters.

Agar models with similar conductivity to real tissue were used, with the electrical conductivity set to the following parameters (σ—Siemens/meter): breast tissue σ = 0.052 and e-tumor tissue σ = 1.125 [33]. Agar models are shaped and sized to fit the standard B-cup electrode brassiere cup (A semi-sphere with measurements 45 mm × 60 mm). Figure 8 illustrates the position and shape of the tumors in the agar models (left model and right model).

Three sizes were used for the emulated tumors (e-tumors):A: Rectangle with dimensions 2 cm × 2.5 cm with a depth of 2.5 cmB: Rectangle with dimensions 1 cm × 2.5 cm with a depth of 1 cm.C: Cylinder with a diameter of 1 cm and a depth of 2.5 cm.

A mold was used to make the agar models, which were built up in steps, first the breast tissue and then the tumor inserts, finishing with a new layer of breast tissue to give the model more stability. Figure 9a,b shows the model after introducing the e-tumors with modified electrical properties. To improve the stability of the model, an additional layer of agar with breast tissue properties is added, as shown in Figure 9c,d of the left and right models, respectively. The final models used to perform the impedance measurements and 3D reconstructions are shown in Figure 9e,f. Red dotted lines were used to highlight the position of the e-tumors within the agar model.

### 2.7. 3D Image Reconstruction

In Electrical Impedance Tomography (EIT) for breast tumor imaging, the process involves identifying and localizing tumors within a reconstructed 3D volume. Tumor locations are determined by calculating x, y, and z coordinates derived from cross-measurements between electrodes. Cross-measurements refer to the electrical impedance values obtained by measuring between various pairs of electrodes placed around the breast, creating a grid of intersecting measurements. These intersecting data points provide a comprehensive view of the impedance distribution within the tissue, allowing for more accurate localization of potential tumors.

The identification of potential tumor sites is based on impedance measurement patterns, with the z coordinate established by analyzing measurements across different vertical levels. This ensures accurate spatial positioning of the tumor within the 3D reconstruction.

The 3D reconstruction process was implemented using MATLAB in conjunction with the EIDORS toolbox, which is specialized for EIT applications. EIDORS enables image reconstruction based on electrical impedance data, playing a critical role in mapping tumor locations in 3D.

Figure 10 describes the workflow of the 3D reconstruction software:

Data acquisition: Impedance measurements are grouped and classified, each measurement corresponding to a set of electrodes. Measurements are sorted by level and by the electrode involved in the measurement.

Filter: A filtering step to eliminate non-contact related measurements; a non-contact measurement refers to an electrode that had no contact with the agar model and output large impedance values. The data is filtered to avoid reconstruction errors due to the high contrast with the measured impedance data.

E-tumor detection: After data filtering, the electrodes that present measurements with values near e-tumors on a regular pattern are identified; these electrodes will be used to identify crossover points in the final reconstruction. The crossover points are areas where the impedance measurements indicate the presence of an e-tumor using the position of the electrodes on the latex cup as a reference.

X,Y calculation: By performing the analysis of the measurements collected at the same electrode level, it is possible to calculate x,y coordinates involving all the identified crossing points.

Z calculation: To determine the Z value of the reconstructed e-tumors, measurements between different electrode levels are analyzed to verify the spread of the tumor at different levels.

Final imaging: With the calculated (x,y,z) values, this set of coordinates is assigned to each e-tumor detected. The e-tumors are reconstructed inside a hemisphere that works as a reference, emulating the position of the electrodes in the latex cup.

## 3. Results

Figure 11 describes the different perspectives used to present the image reconstruction results in this paper. The electrodes in each cup have been numbered (from E1 to E24). The electrode numbering starts from the inner ring array at the top electrode (E1), continues clockwise until all electrodes in the ring array are completed, and continues in the same numbering order until all 24 electrodes are completed (E24).

Figure 12 shows the top view of the 3D reconstruction of the agar molds (left model and right model). The green circles represent the position of the electrodes on the latex cup. A cm reference system was used to facilitate the location of the localized e-tumors. Figure 10a shows e-tumor A in the upper part (between E8 and E1), which stands out from the others because of its larger size. On the left (in front of E7) and right (in front of E3) sides of the reconstruction, e-tumor B is shown, and in the lower part, e-tumor C (in front of E5) is shown.

Analyzing Figure 12b, in the upper central part is the e-tumor B (in front of E1), in the upper part of the left (in front of E8) and right (in front of E2) sides are the e-tumors C, in the lower central part (between E4 and E6) is the e-tumor A, which has the largest size in this reconstruction. One of the drawbacks of analyzing cross-sectional views (2D views of a plane) in EIT is that they do not provide sufficient information for propagation analysis (presence of an e-tumor in more than one plane). The present paper discusses one of the advantages of 3D reconstruction in the analysis of the propagation of e-tumors C, using an additional view to the top view.

Additional views are presented that provide more information about the geometric location of the e-tumors. They provide a visual aid for pre-diagnosis and the location of the tumors in relation to the coordinates and the distance between the electrode positions. For this experiment, it is observed that the emulated tumors (e-tumors) with a depth greater than 2 cm are represented in two levels of the reconstruction; the A,C e-tumors of the left and right models are reconstructed in the two lower levels (level 2 and level 3) of the 3D reconstruction, unlike the e-tumors in both models which, due to their depth of 1 cm, are only reconstructed in the middle level (level 2).

In the frontal view of the left model (Figure 13a), it is observed that e-tumor B is in level 2, and the larger e-tumor A (in front of E13 and E21) spreads between levels 2 and 3. Another view of the model should be analyzed in order to differentiate e-tumor A from e-tumor C of left agar models. Figure 13b shows e-tumors C and A (in front of E13 and E21) propagating between level 2 and level 3; the top view is required to be analyzed to differentiate e-tumor B from e-tumor C.

The isometric view (Figure 14) is also presented, which is useful as a starting point to identify the e-tumors detected in the reconstruction. It allows us to assess the spread of the e-tumor (by analyzing other views) within the breast model. Visually, a 3D reconstruction allows the localization of the e-tumors by using the location of the electrodes as a reference.

## 4. Discussion

The development and implementation of a control system and a wearable brassiere cup design for tumor detection in agar models using electrical impedance tomography (EIT) represent significant advances in the field of medical imaging. The control system developed demonstrates a modular design that facilitates adaptability and customization for different medical testing environments. Using a multiplexer for power distribution and a state machine for efficient data acquisition, the system optimizes the scanning process. The integration of an Arduino mega 2560 into a double-layer PCB streamlines the control mechanism and increases the overall functionality and portability of the system.

The introduction of a modified brassiere cup device enhances the portability and usability of the EIT system, providing a patient-friendly solution for tumor detection. The integration of the electrodes into a standard B-cup brassiere, along with a Velcro inner liner, ensures easy attachment and comfort for the wearer. This innovation opens the door to non-invasive and accessible imaging techniques, especially in remote or underserved areas.

The use of MATLAB software (R2019a (9.6.0.1072779)) with the EIDORS toolbox enables efficient 3D image reconstruction based on impedance measurements. The reconstructed images provide valuable insight into the spatial distribution of tumors within the agar models. By visualizing the electrode positions and tumor locations from multiple perspectives, researchers can effectively analyze and interpret the imaging results, aiding in tumor localization and diagnosis. However, 3D reconstructions suffer from one of the major limitations of electrical impedance tomography, which is low imaging resolution.

While the current study shows promising results in tumor detection using EIT, there are several avenues for future research and development. Further refinement of the control system and the design of the wearable brassiere cup could improve usability and scalability, potentially leading to clinical application. In addition, continuous advances in image reconstruction algorithms and modeling techniques may improve the accuracy and resolution of EIT imaging, thereby expanding its utility in medical diagnostics.

The integration of a versatile control system, wearable brassiere cup design, agar models, and advanced image reconstruction techniques represents a significant step forward in the application of EIT for tumor detection. By addressing the challenges of portability, usability, and imaging accuracy, this study contributes to future innovations in medical imaging technology.

Future work requires clinical trials that utilize patient feedback and system failure detection to suggest improvements or modifications to the current prototype, if necessary.

## 5. Conclusions

The project aims to conduct future clinical trials to demonstrate the feasibility of prediagnosis of tumor breast, with a focus on communities or locations that do not have the medical care for standard mammography diagnosis. The modularity allows easy modification of the prototype according to the characteristics required for the tests. It is possible to increase the number of electrodes to improve the resolution of the prototype, although this would increase the cost of computer processing and require more hardware. One of the advantages of using electrical impedance tomography is the ease and safety of testing, although the main disadvantage will be lower resolution. Clinical studies (future work) are needed to make modifications to the method of acquiring impedance measurements, aesthetic adjustments such as cup size selection, or adaptation of the wearable bra cup design. An important advantage of this study was the use of agar and e-tumor models to facilitate the experimental phase, which facilitated the identification of errors in the measurement method and in the 3D reconstruction software. The 3D reconstruction allowed the identification of the propagation (an e-tumor at more than one level), as opposed to a 2D cross-sectional view widely used in electrical impedance tomography. There are potential areas to improve the project, including (but not limited to) the use of technologies such as 3D printing to improve the ergonomics of the designed brassiere cup, focusing on the use of a compact (rather than modular) architecture to reduce the size of the system, the use of artificial intelligence to reduce the impact of the low resolution of the electrical impedance tomography; Utilizing deep learning algorithms has shown promising results. These networks, such as convolutional neural networks (CNNs), enhance the reconstruction accuracy by learning complex patterns from training data. Applying machine learning techniques to EIT involves training models to optimize reconstruction based on various factors, contributing to better image quality. 

## Figures and Tables

**Figure 1 sensors-24-06370-f001:**
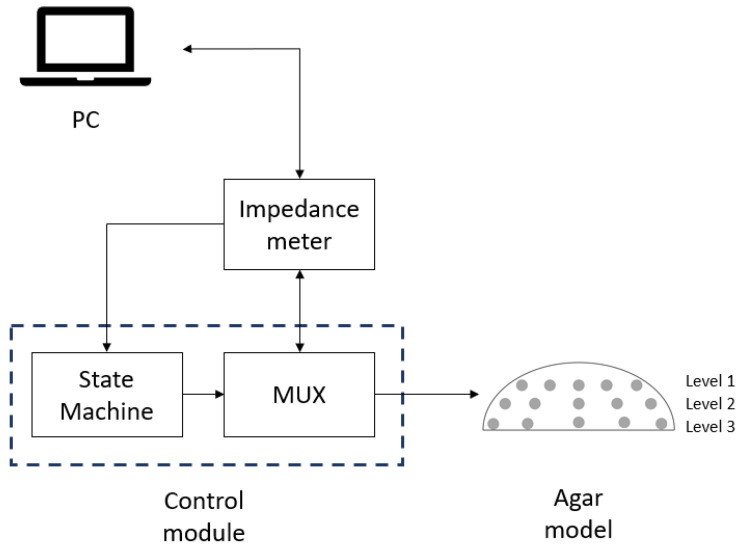
Proposed system diagram for tumor detection in agar models.

**Figure 2 sensors-24-06370-f002:**
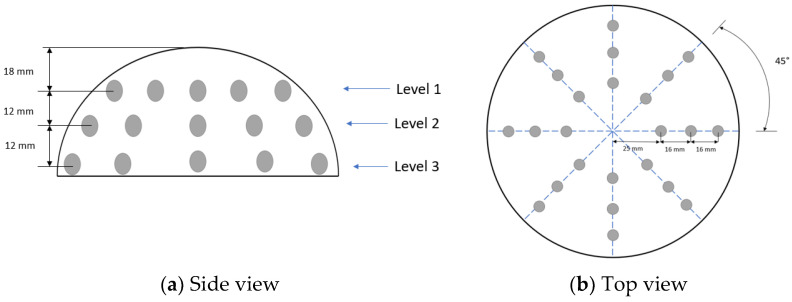
Electrode distribution in latex cup.

**Figure 3 sensors-24-06370-f003:**
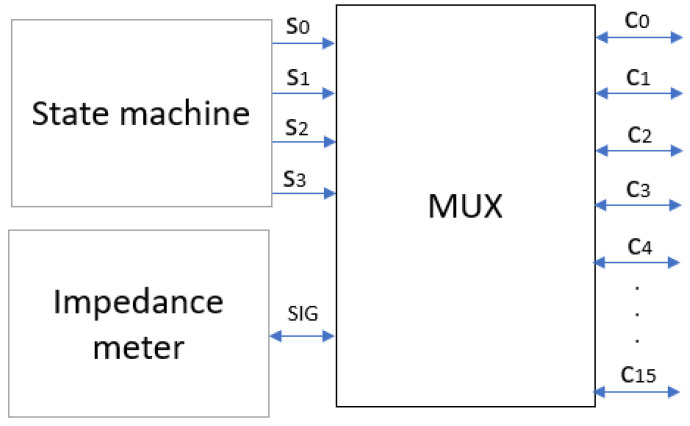
MUX as a signal distributor.

**Figure 4 sensors-24-06370-f004:**
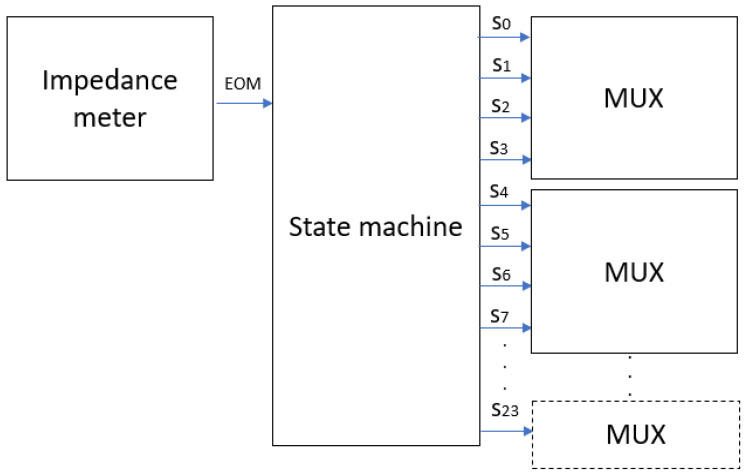
State machine connection.

**Figure 5 sensors-24-06370-f005:**
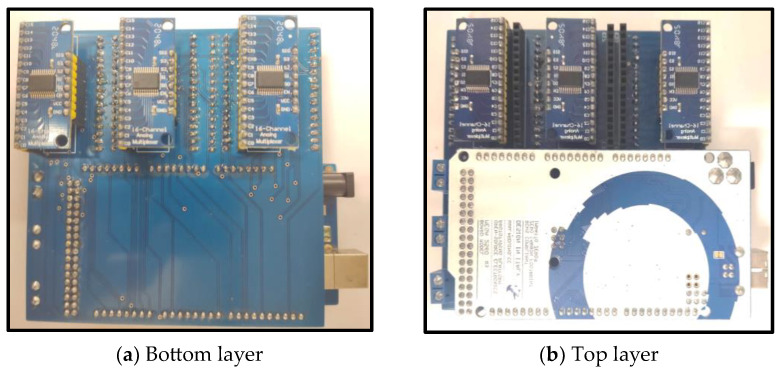
Control module.

**Figure 6 sensors-24-06370-f006:**
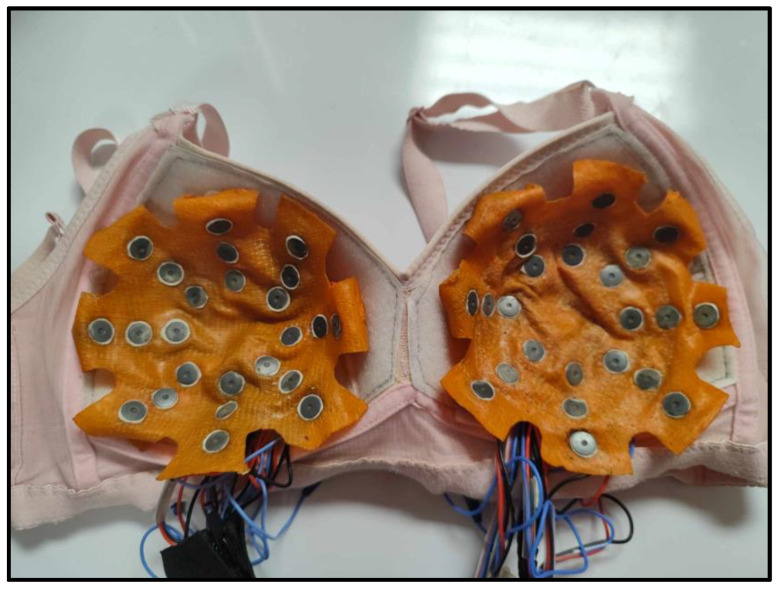
Electrodes mounted on brassiere cup.

**Figure 7 sensors-24-06370-f007:**
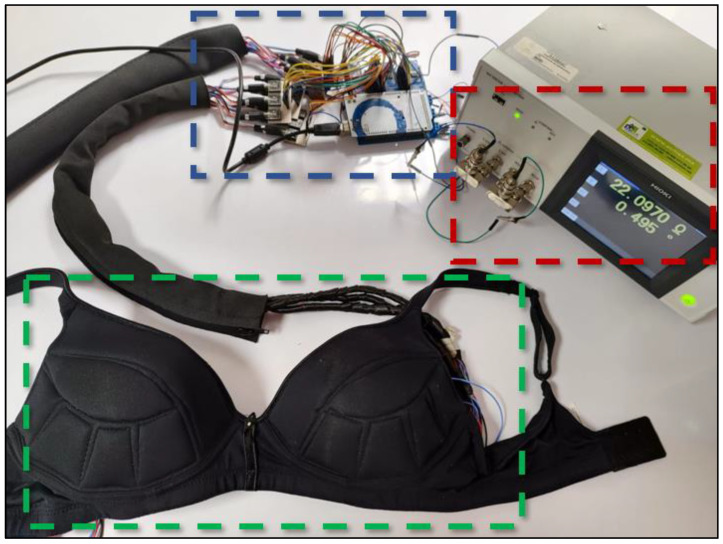
Prototype ready for impedance measurement. Impedance meter (red color frame), control module (blue color frame) and the brassiere cup (green color frame).

**Figure 8 sensors-24-06370-f008:**
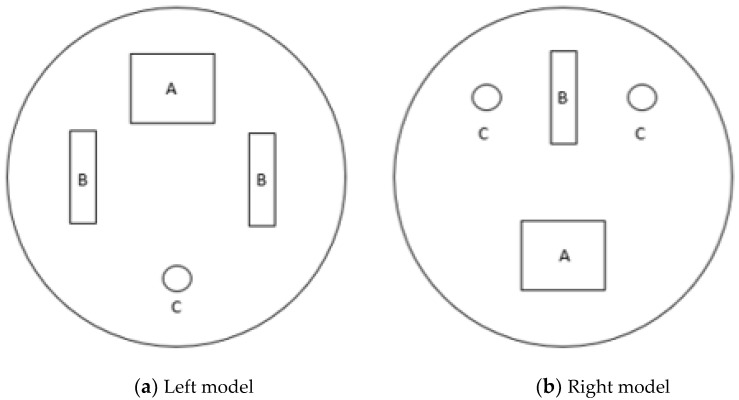
Agar models representing female breasts with e-tumors.

**Figure 9 sensors-24-06370-f009:**
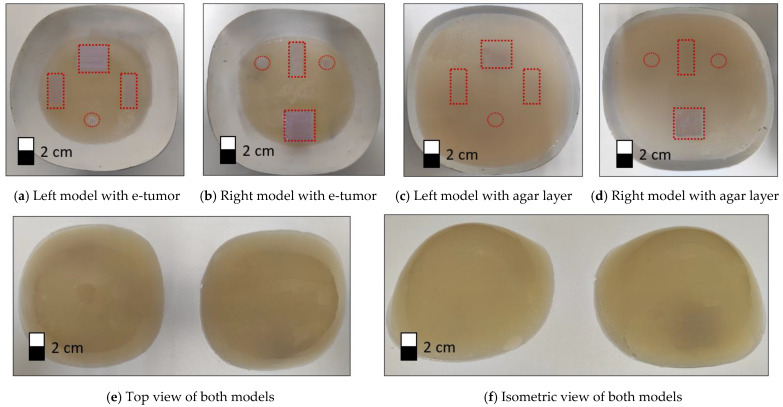
Agar models. Dotted frames indicate the position of the e-tumors in the agar model.

**Figure 10 sensors-24-06370-f010:**
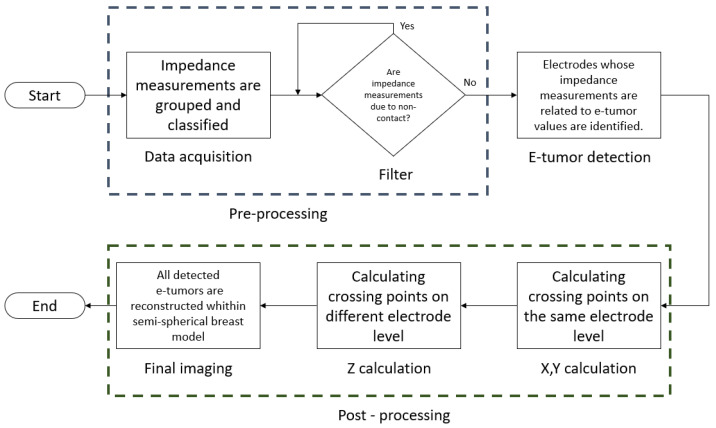
3D reconstruction diagram.

**Figure 11 sensors-24-06370-f011:**
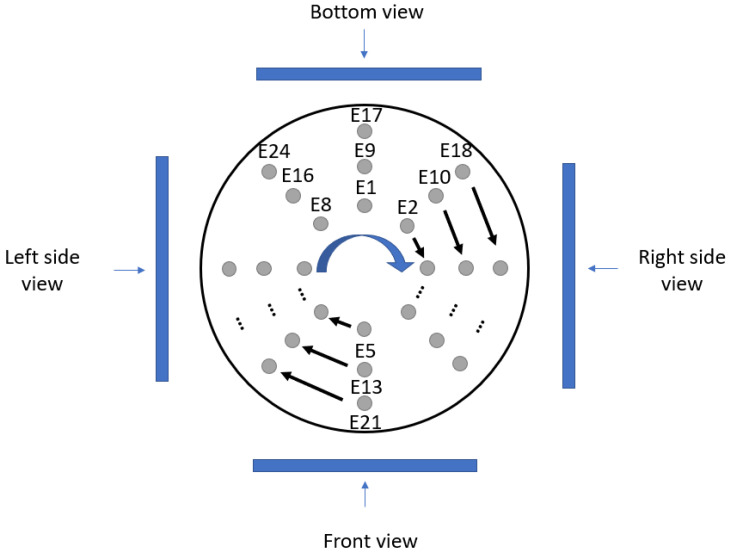
Different views of the 3D reconstruction (from top view).

**Figure 12 sensors-24-06370-f012:**
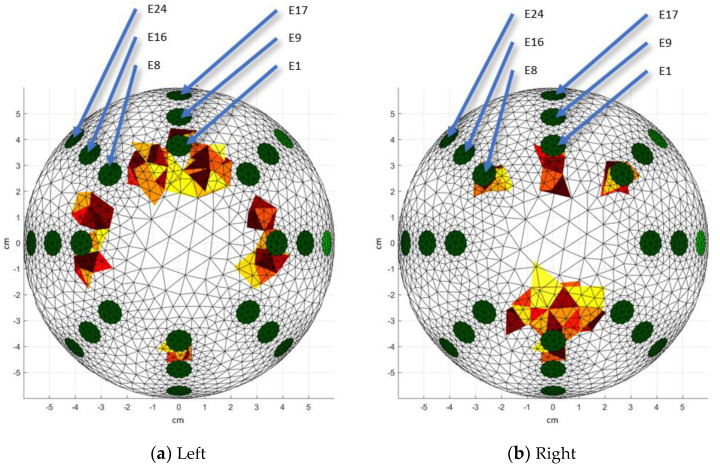
3D reconstructions: Top view. Red colors represent the e-tumors, green colors represent the position of the electrodes. Warm-toned color scales (such as red, yellow, and orange) are commonly used in 3D visualizations to represent the e-tumor, enhancing the clarity of spatial representation. These colors are selected due to their association with higher intensity regions, making them suitable for highlighting areas of interest.

**Figure 13 sensors-24-06370-f013:**
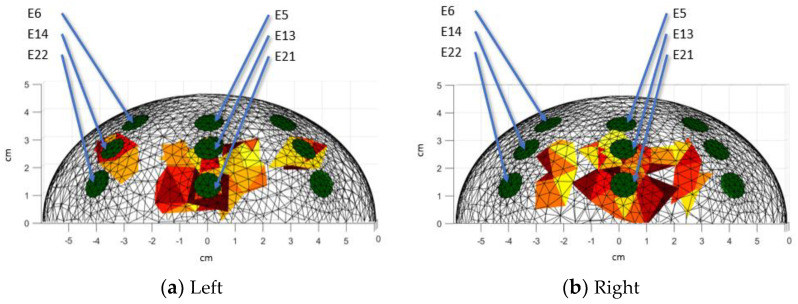
3D reconstructions: Front view. Red colors represent the e-tumors, green colors represent the position of the electrodes. Warm-toned color scales (such as red, yellow, and orange) are commonly used in 3D visualizations to represent the e-tumor, enhancing the clarity of spatial representation. These colors are selected due to their association with higher intensity regions, making them suitable for highlighting areas of interest.

**Figure 14 sensors-24-06370-f014:**
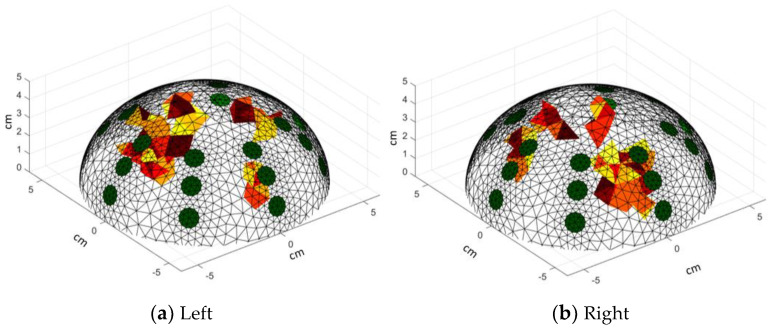
3D reconstructions: Isometric view. Red colors represent the e-tumors, green colors represent the position of the electrodes. Warm-toned color scales (such as red, yellow, and orange) are commonly used in 3D visualizations to represent the e-tumor, enhancing the clarity of spatial representation. These colors are selected due to their association with higher intensity regions, making them suitable for highlighting areas of interest.

## Data Availability

Data are contained within the article.

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
