# Peer review of "Modular and Portable System Design for 3D Imaging of Breast Tumors Using Electrical Impedance Tomography"

_sensors, 2024, doi:10.3390/s24196370_

Round 1

Reviewer 1 Report

Comments and Suggestions for Authors

The work presents a prototype of electrical impedance tomography (EIT)-based method for breast tumor detection. The authors hoped to further develop a portable and affordable system in clinical applications for less developed regions. Although the mechanism of EIT-based method isn’t new, the research is still sound in preliminary check of breast tumors.

There are some points to be addressed in this work:

1.      The authors only presented a diagram of the system, but also need show the entire real equipment in the paper since they have measured the data.

2.      Due to experimental limit, agar models were used to mimic the real tissues while emulating tumors were embedded into the agar gel. The introduction for the e-tumors was lacking, and need be told how the material mimics breast tumors.

3.      Figures 10-12 showed the 3D reconstructions of measured EIT signals. However, the measured raw data need be presented as samples in the paper as well, and how the 3D reconstructions were processed from the EIT signals are to be introduced.

4.      The Abstract need be re-organized. For instances, a) agar model need be mentioned before the preliminary testing; b) the second sentence wasn’t accurate as no involvement of real tissue in this study; c) the words “accurate diagnosis and treatment plan” were overstated since this design was just a prototype and the measured data were crude far from clinical diagnosis, whereas preliminary screening sounds more appropriate.

5.      The pictures on Figure 8 need add size-scale bars.

Author Response

We would like to express our sincere gratitude for your time and effort in reviewing our manuscript. We greatly appreciate your valuable comments and suggestions, as they have helped us improve the quality of our work.

We have carefully considered each of your recommendations and have made the necessary revisions to address the concerns raised. In the revised manuscript, we have provided detailed explanations of the changes made and how they align with your feedback. We believe these updates have strengthened our contribution, and we hope the revised version meets your expectations.

Thank you once again for your insightful suggestions and for providing us the opportunity to enhance our work.

Reviewer 2 Report

Comments and Suggestions for Authors

The introduction section is poorly structured, lacking critical analysis, novelty, and a coherent flow of information. It does not highlight the unique contribution of the work, making it difficult for readers to understand what sets it apart from existing research. Furthermore, the quality of the graphs is substandard, reducing the clarity of the data presented. Author may incorporate more details about tumor and its properties https://ieeexplore.ieee.org/abstract/document/9063185 and nature of cancer. Fig. 5(b) is not readable, completely blur. It is also needed to discuss about breast tissue property and breast tumor Discussing the dielectric contrast between healthy and cancerous tissues allows for better sensitivity and specificity in detecting abnormalities. https://ieeexplore.ieee.org/abstract/document/9036481. Why authors choose to apply agar model? Is there any specific reason? Remove patent information from the paper. No need to incorporate it within the manuscript. Authors are recommended to discuss about microwave imaging method to detect human breast  and mention some comparative remarks within the text. Authors are encouraged to improve the introduction by providing a well-structured comparison with existing works, clearly emphasizing the advancements made, and discussing how this work addresses gaps in the current literature. How author can say their proposed system shows better results than existing systems?  

Comments on the Quality of English Language

moderate

Author Response

(The authors gave the same response as above.)

Round 2

Reviewer 2 Report

Comments and Suggestions for Authors

Authors have addressed all the concerns properly. Now it may be considered for publication.

Comments on the Quality of English Language

acceptable